# Chronic hepatitis B virus case-finding in UK populations born abroad in intermediate or high endemicity countries: an economic evaluation

Natasha K Martin,[1,2] Peter Vickerman,[2] Salim Khakoo,[3] Anjan Ghosh,[4] Mary Ramsay,[5,6] M Hickman,[2] Jack Williams,[7] Alec Miners[7]

[1]Department of Medicine, University of California San Diego, La Jolla, California, USA
[2]Population Health Sciences, University of Bristol, Bristol, UK
[3]Clinical and Experimental Sciences, University of Southampton, Southampton, UK
[4]NHS London Borough of Bexley, London, UK
[5]Immunisation, Public Health England, London, UK
[6]Epidemiology, London School of Hygiene and Tropical Medicine, London, UK
[7]Public Health and Policy, London School of Hygiene and Tropical Medicine, London, UK

**Correspondence to**
Alec Miners;
alec.miners@lshtm.ac.uk

## ABSTRACT

**Objectives** The majority (>90%) of new or undiagnosed cases of hepatitis B virus (HBV) in the UK are among individuals born in countries with intermediate or high prevalence levels (≥2%). We evaluate the cost-effectiveness of increased HBV case-finding among UK migrant populations, based on a one-time opt out case-finding approach in a primary care setting.

**Design** Cost-effectiveness evaluation. A decision model based on a Markov approach was built to assess the progression of HBV infection with and without treatment as a result of case-finding. The model parameters, including the cost and effects of case-finding and treatment, were estimated from the literature. All costs were expressed in 2017/2018 British Pounds (GBPs) and health outcomes as quality-adjusted life-years (QALYs).

**Intervention** Hepatitis B virus case-finding among UK migrant populations born in countries with intermediate or high prevalence levels (≥2%) in a primary care setting compared with no intervention (background testing).

**Results** At a 2% hepatitis B surface antigen (HBsAg) prevalence, the case-finding intervention led to a mean incremental cost-effectiveness ratio of £13 625 per QALY gained which was 87% and 98% likely of being cost-effective at willingness to pay (WTP) thresholds of £20 000 and £30 000 per additional QALY, respectively. Sensitivity analyses indicated that the intervention would remain cost-effective under a £20 000 WTP threshold as long as HBsAg prevalence among the migrant population is at least 1%. However, the results were sensitive to a number of parameters, especially the time horizon and probability of treatment uptake.

**Conclusions** HBV case-finding using a one-time opt out approach in primary care settings is very likely to be cost-effective among UK migrant populations with HBsAg prevalence ≥1% if the WTP for an additional QALY is around £20 000.

## INTRODUCTION

Worldwide, the burden of liver disease continues to rise and remains an urgent public health problem.[1] It is estimated that viral hepatitis is in the top 10 leading causes of mortality globally,[2] the majority due to infection with hepatitis B virus (HBV).[3] Chronic infection with HBV can lead to liver fibrosis, cirrhosis, hepatocellular carcinoma (HCC) and death in the absence of treatment. It is estimated that over 5% of the world's population are chronic carriers of HBV.[4] Globally, HBV burden is highest in low-middle income countries in areas such as Sub-Saharan African and East Asia.[3] HBV is spread through exposure to infected blood or body fluids, with the majority of chronic infections acquired perinatally or during childhood.[1] Recently, effective antiviral treatment for HBV has become available which may achieve long-term viral suppression and slow progression of disease.[5 6]

The UK has a low burden of HBV, with an estimated 0.4% of adults infected with chronic hepatitis B (CHB),[7] and only approximately 320 cases of acute HBV reported in England in 2015.[8] The vast majority (80%–90%) of newly diagnosed chronic HBV infections are among migrant individuals living in the UK that were

## Strengths and limitations of this study

► Our cost-effectiveness evaluation is one of few studies evaluating hepatitis B virus case-finding among populations born abroad in intermediate to high endemicity countries.
► Strengths include numerous sensitivity analyses assessing how cost-effectiveness varies for a range of different prevalences, intervention effect and cost, thus increasing the generalisability of our results to other similar interventions and different settings.
► A key limitation is uncertainty in the exact cost or effect of this intervention if scaled up to a national level.
► The model, due to a lack of available data, did not incorporate any additional impact of household contact tracing of diagnosed cases.
► The model also does not incorporate the possibility of simultaneous testing for hepatitis C virus.

born overseas in countries with intermediate (2%–7%) or high HBV prevalence (≥8%) as defined by the WHO,[9] such as China or Pakistan.[10–12] Although uncertain, it is also likely that a considerable number of people with chronic HBV remain undiagnosed. For example, in one study in Bristol only 12% of migrants born in countries with endemic prevalence >2% had been tested for HBV.[10] Due to the often asymptomatic nature of chronic infection,[13] individuals with HBV infection can often remain undiagnosed until they develop advanced liver disease. It is critical, therefore, that increased case-finding among UK migrant populations is enhanced to ensure timely treatment and follow-up to prevent complications from liver disease.

The UK, like many countries worldwide, recommends universal screening of pregnant women to identify and immunise neonates exposed to HBV infection, which has been shown to be highly cost-effective and under some circumstances cost-saving.[14] However, the UK is one of only six countries in Europe which does not offer universal immunisation against hepatitis B (along with Denmark, Finland, Iceland, Norway and Sweden). These countries have a very low HBV endemicity and so it is unlikely to be cost-effective to introduce a separate universal HBV vaccination programme.[15] Recent assessments of the cost-effectiveness of universal childhood HBV vaccination suggest that it may be cost-effective if introduced with other vaccines as a component of a hexavalent vaccine—the UK moved to such a product in 2017.[15] Nonetheless, infant vaccination is unlikely to have a great impact on the prevalence of chronic HBV in countries such as the UK because few transmissions are thought to occur once people have entered the country.[16] For these reasons, there remains a critically important role for case-finding activities.

While studies in The Netherlands have shown the cost-effectiveness of one-time screening programmes (where a test offer is mailed to migrant individuals identified through a population registry[17]), until recently there has not been a published evaluation from a UK perspective. This changed earlier this year when the results of a randomised controlled trial (HepFREE) showed that incentivised screening of HBV and hepatitis C virus (HCV) in first-generation and second-generation migrants in a primary care setting was shown to be effective and cost-effective in the UK; the incentive included a startup payment of £500 per general practice (GP), £25 for each enrolled participant and support from a dedicated clinician 3 days a week.[18] However, in contrast to an incentivised screening approach, pilot data from the UK also indicate that an opt-out HBV case-finding approach in primary care settings without incentives was also highly effective, and potentially a less expensive approach.[19] Additionally, it was unclear in the previous analysis for the HepFREE trial how much the cost-effectiveness was driven by HCV versus HBV outcomes, and whether the intervention was cost-effective for HBV alone. Further, it is unknown how the cost-effectiveness of HBV case-finding

could vary for a range of prevalences (which likely vary by country of origin), costs and uptake rates that may occur when the interventions are rolled out across different settings.

The aim of this paper is to evaluate the cost-effectiveness of increased HBV case-finding among UK migrant populations born in intermediate or high endemicity countries, based on a one-time opt out case-finding approach in primary care settings. Importantly, to increase the generalisability of our results to other similar interventions and different settings, we assess how the cost-effectiveness of HBV case-finding varies for a range of different prevalences, intervention effect and cost.

## METHODS

The economic evaluation was undertaken using a Markov approach, where a closed cohort of UK individuals born in countries with intermediate or high prevalence levels (≥2%) move between a set of discrete health states representing HBV infection stage.[20 21] A UK National Health Service's cost perspective was used. All costs were displayed in British Pounds (GBPs), 2017/2018 prices and a 40-year time horizon was used with an annual time step. Health outcomes were expressed in terms of quality-adjusted life-years (QALYs). QALYs and costs were discounted at 3.5% per annum according to UK National Institute for Health and Care Excellence (NICE) recommendations.[22] Uncertainty in the results was examined using deterministic and probabilistic sensitivity analysis (PSA); distributions shown in the tables relate to the PSA analysis. Each PSA consisted of 5000 runs. HBV transmission was not included in the model as most infections are likely to occur in UK migrant populations before entering the UK.[16]

### Intervention and target population

A systematic literature review found few studies evaluating HBV case-finding in migrant or other high-risk populations, nor have many studies been published since this review.[18 23] Our study evaluates the cost-effectiveness of HBV case-finding in the UK for individuals born in countries with intermediate or high prevalence levels (≥2%). The base case analysis uses the results from an uncontrolled study in which Pakistani/British Pakistani people registered at GPs in London's East End were written to and invited to 'opt out' of being tested for hepatitis B and C infection. Those who did not opt out were telephoned and asked to attend a clinic for testing.[19] The intervention was designed to increase the likelihood of testing for each infection, assumed in this analysis to occur over the initial model cycle of 1 year. After this time, the intervention effect was assumed to be zero, with the probability of testing reverting to background levels. The comparator programme or 'no intervention' was defined as the background likelihood of testing through existing routes such as sexual health or genitourinary medicine clinics, antenatal clinics or primary care.[24] Although we base our

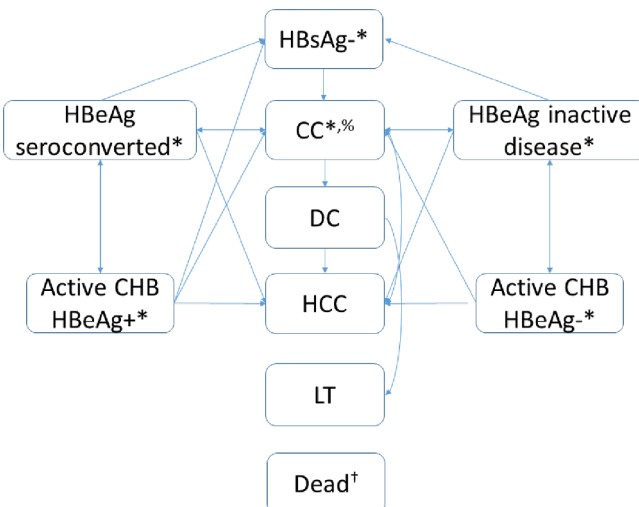

**Figure 1** HBV model schematic. The arrows denote possible transitions between states. CC, compensated cirrhosis; CHB, chronic hepatitis B virus; DC, decompensated cirrhosis; HBeAg, hepatitis b virus e antigen; HBsAg, hepatitis B virus surface antigen; HCC, hepatocellular carcinoma; LT, liver transplant; *individuals may or may not know their infection status. %Individuals with CC responding to treatment were assumed to keep the costs and utility associated with CC, but with disease progression probabilities equivalent to HBeAg seroconversion/inactive disease. †Transitions permitted from all health states to death.

analysis on data from a study among Pakistani/British Pakistani individuals in London, we evaluate the potential impact of this intervention in populations with a range of HBV prevalences as observed among UK migrants born in countries with intermediate or high prevalence levels (≥ 2%).

## Model structure

The Markov model was created to represent HBV disease progression and current understanding of policies regarding disease management (figure 1). The natural history element of the model was largely based on a model developed by Shepherd et al.[25 26] The model simulates a cohort of people, a proportion of whom are positive for hepatitis B surface antigen (HBsAg+). For this analyses, we refer to 'HBV prevalence' as the proportion of individuals who are HBsAg+. Individuals who are negative for HBsAg- remain in the model with a general population level of mortality but incurring no HBV-related costs, other than the possibility of being tested for infection. Known HBsAg+ people were assumed to undergo a full viral profile when initially diagnosed. Acute HBV infection was not included in the model as it is likely that people would have been infected much longer than 6 months ago.

Among HBsAg+ individuals, the model stratifies by mutually exclusive stages of CHB, including HBeAg seroconverted (where ALT (alanine aminotransferase) levels and HBV DNA are both low), active CHB hepatitis B e-antigen positive (HBeAg+) disease, active CHB hepatitis B

e-antigen negative (HBeAg-) disease and inactive CHB HBeAg- (where ALT levels and HBV DNA are both low). Individuals progressed from CHB to compensated cirrhosis, decompensated cirrhosis (DC), HCC, liver transplant and post-transplant stages if appropriate drug treatment was not initiated or failed. Due to the severity of the disease and likely presentation, the infection status of all individuals with CHB was assumed to become known when they developed DC, HCC or required a liver transplant. Individuals could die from non-HBV related causes from any health state.

Individuals who had raised ALT and HBV (active) levels and who were CHB HBeAg+ were assessed for fibrosis and offered treatment with pegylated interferon for the first year, followed by tenofovir until seroconversion is achieved (as per NICE guidelines[27]) or later stage CHB developed. We assumed successful treatment of these individuals resulted in normalisation of ALT and lowering of HBV DNA levels, therefore resulting in transition to the HBeAg seroconverted stage. Individuals with no evidence of compensated cirrhosis stopped treatment at this time.[27] Individuals with active CHB who were HBeAg- also received pegylated interferon for the first year, followed by tenofovir if they had not developed inactive CHB HBeAg- disease.[27] However, even following the development of inactive disease, they were assumed to stay on treatment indefinitely to sustain the achieved level of viral suppression.[27] Individuals with evidence of compensated cirrhosis were assumed to remain on tenofovir as long as no further disease progression was recorded, irrespective of e-antigen status.[27] All individuals were assumed to stop treatment on progression to DC or later stages of disease.

Individuals with CHB whose infection status was unknown and those that tested HBsAg+, but declined treatment, were assumed to develop progressive disease according to a set of defined transition probabilities, with different probabilities used for those who accepted treatment (online supplementary tables 1 and 2). As the focus of this analysis is on case-finding, we do not model possible adverse events associated with treatment or treatment resistance.

## Model parameters
### HBV prevalence among migrant populations to the UK

There is substantial heterogeneity in HBV burden between different migrant populations in the UK depending on their country of origin. Additionally, HBV prevalence among UK migrants may be different compared with their country of origin; a recent UK study of antenatal testing showed the prevalence in migrants was generally less than published estimates for the country of origin, with only Eastern Asia having a higher than expected prevalence.[11] Public Health England (PHE) data on those undergoing routine diagnostic testing suggest that the HBV prevalence among all Asian or British Asian people in the UK is approximately 2%; however, these data do not specify country of origin in any further detail.[28]

By contrast, the HBV prevalence estimates obtained through targeted studies or antenatal testing have identified a range of prevalence among UK migrants born in countries with intermediate-high HBV endemicity, such as 17% (Vietnam-born), 7%–10%[29 30] (China-born), 3%–6% (Somalia-born), 1%–3% (Pakistan-born), 0.5%–1.5% (Bangladesh-born), 0.7% (Poland-born) and 0.5% (India-born).[16 31–33] The recent HepFREE trial found a lower prevalence of 1.1%, varying by country of origin, although this included second-generation migrants that were born in the UK.[18]

Due to the uncertainty in prevalence within populations, and the likely wide variation between populations, in the base case, we assume an HBV prevalence (HBsAg+) of 2%, but explore a range of values (from 0.05% to 10%) in the sensitivity analysis.

### Transition probabilities

Transition probability values, representing the likelihood of moving between health states, for untreated disease stages were based on those reported in a 2006 UK Health Technology Assessment report (online supplementary tables 1 and 2).[25]

### Background testing rate and diagnostic accuracy

The background rate of testing for migrants in the absence of the intervention was estimated using data from PHE, indicating a probability of 2.6% per year.[24] The HBsAg diagnostic test was assumed to be 100% accurate.

### Referral and treatment effect

Few studies have quantified the number of people diagnosed with CHB who are subsequently referred to, and accept, appropriate further clinical investigations for their infection. However, interruptions in the cascade of care postdiagnosis are known to be an issue in the management of CHB and HCV infection both in the UK and elsewhere, particularly in migrant populations.[34] We therefore include a single probability of being referred for specialist care following a HBsAg+ test result, attending the appointment and starting treatment for those eligible. In the absence of HBV-related data, we use data on the proportion of individuals who were identified using algorithmic approaches as being Asian and who tested positive and subsequently received treatment for chronic HCV from 2004 to 2015 (0.42, based on data supplied by PHE, personal communication with PHE staff). However, we consider this parameter to be highly uncertain and undertake sensitivity analysis around it using a wide range of alternative values (10% to 60%).

While a systematic review and meta-analysis of the effects of drug therapy for CHB is available,[35] we estimated the impact of antiviral treatment using data from a study which has a much longer follow-up period (5 years rather than 1 year).[36] For HBeAg+ individuals, we assumed 20% would e-antigen seroconvert after 1 year of treatment with pegylated interferon and 5.4%/year following treatment with tenofovir, resulting in 40% having seroconverted by 5 years. For HBeAg- individuals, we assumed a 75% probability of response (development of inactive disease) following the initial 1 year of pegylated interferon and 2.3%/year following treatment with tenofovir. Therefore, we assumed that 84% would develop inactive disease by 5 years. Irrespective of whether individuals were HBeAg+ or HBeAg-, they were assumed to continue treatment after 5 year with tenofovir until they responded to it assuming the same constant rate of response.

The probability of responding to treatment was assumed to be the same for people with or without compensated disease. However, once people developed compensated disease, it was assumed not to regress following treatment, and the costs and disutility associated with it would remain. The only benefit of treatment in this group was slower progression to poorer health states compared with not being treated.

### Intervention effect

The base case probability of testing for HBsAg in the intervention arm was based on a one-time 'opt out' option within a GP setting; 223 out of 1134 (19.7%) eligible tested after being identified using a GP registries database and responding to a written invite.[19]

### Cohort demographics and initial stage distribution

PHE data suggest that the average age at HBV diagnosis in the UK Asian population is approximately 35 years of age,[28] which we use as the base-case starting age in our model but vary in the sensitivity analysis. The proportion of people with CHB who were HBeAg+ in our starting cohort was assumed to be 0.14 ((71/490) personal communication with PHE staff). The proportion of people who had seroconverted, or developed inactive disease, before being tested for HBsAg, was assumed to be 80% (personal communication with PHE staff). It was further assumed that 44% of people with active HBeAg+ or HBeAg- disease had already developed compensated cirrhosis.[37]

### Health utilities and costs

Utility values related to HBV infection were sourced from the review by Shepherd *et al*[25] and Takeda *et al*[26] (online supplementary table 3). The costs of HBV testing/monitoring, antiviral treatment and health-state specific costs were taken from a number of published sources[25 37] (table 1), inflated to GBP £2017 where appropriate using the NHS Hospital and Community Health Services Pay and Prices Index and the Health Service Cost Index.[38 39] The intervention cost was estimated at £4 per person eligible for testing. This cost relates to the resources required to identify and invite each individual for a test and excludes the cost of any tests and treatments. Thus, if 100 individuals were eligible for testing, the total cost of the intervention was £400 irrespective of how many people attended for a test. The importance of this assumption

**Table 1** Annual costs in 2017/2018 UK prices (£)

| Cost | Mean | 95% interval of sampled range* | Source |
|---|---|---|---|
| Intervention cost per person eligible for testing† | 4 | – | Assumption |
| HBsAg test (laboratory) | 10 | – | Assumption |
| Pegylated interferon | 3979 | – | BNF[45] |
| Tenofovir | 2453 | – | BNF[45] |
| ALT and ultrasound | 77 | – | Assumption[45] |
| Full viral profile | 432 | – | Assumption[45] |
| HBeAg+ seroconverted/HBeAg- ALT/DNA low* | 335 | 240–446 | Shepherd[25] |
| HBeAg+/HBeAg active disease§ | 674 | 480–896 | Shepherd[25] |
| Compensated cirrhosis | 1606 | 1052–2283 | Crossan[37] |
| Decompensated cirrhosis | 38 212 | 21 848–60 645 | Crossan[37] |
| Hepatocellular carcinoma | 38 212 | 21 848–60 645 | Crossan[37] |
| Liver transplant (first year) | 67 698 | 57 301–79 287 | Crossan[37] |
| Liver transplant (subsequent years) | 17 231 | 5415–35 399 | Crossan[37] |

*Sampled values from the probabilistic sensitivity analysis using a gamma distribution.
†One off cost.
§Costs are additional to*.
BNF, British National Formulary; HBeAg, hepatitis b virus antigen; HBsAg, hepatitis B virus surface antigen.

was assessed in the sensitivity analysis given the extent of uncertainty.

## Main outcomes

Our main results incorporate a PSA, in which relevant parameters are simultaneously sampled 5000 times to represent underlying uncertainty, including the costs, utilities, probabilities and disease progression parameters. We present total and incremental costs, QALYs and incremental cost-effectiveness ratios (ICERs). Mean and 2.5%–97.5% centile (95% CI) results are presented. We additionally present the proportion of simulations which are cost-effective under £20 000 and £30 000 willingness to pay (WTP) per additional QALY thresholds.

## Sensitivity analyses

To test the robustness of the results to alternative assumptions, we undertook extensive one-way sensitivity analyses on starting age, discount rate, drug cost, time horizon, treatment uptake, intervention effect and intervention cost. Finally, due to the uncertainty surrounding the intervention cost and impact if scaled-up to the national level and among different migrant populations, we undertook a threshold analysis where we evaluated the minimum HBV prevalence at which the intervention remains cost-effective at a WTP threshold of <£20 000 per QALY gained with varying intervention cost (between £1 and £20, £4 per person eligible at base-case), intervention effect (between 5% and 30%, 19.7% uptake at base-case) and HBsAg prevalence (between 1% and 10%, 2% base-case). We displayed the results of this sensitivity analysis as a contour map.

## RESULTS

### Base-case 2% HBsAg prevalence

At a 2% HBsAg prevalence, the HBV case-finding intervention resulted in mean incremental costs and QALYs of about £28 and 0.002, respectively, over the 5000 samples, corresponding to an ICER of £13 625 per QALY gained (95% CI £7121–£27 588). The intervention was 87% and 98% likely to be cost-effective at £20 000 and £30 000 WTP per additional QALY thresholds, respectively (online supplementary figure 1). Most of the univariate sensitivity analyses produced ICERs below a £20 000 WTP threshold (figure 2), including reducing the likelihood of testing from 19.7% to 5% (£19 323/ QALY gained). However, the exceptions were assuming a 20-year time horizon instead of 40 years (£22 713/ QALY gained), discounting QALYs at 6% instead of 3.5% (£21 970/QALY gained), not discounting costs instead of 6% (21 521/QALY gained) and doubling the costs of all drug treatments from £3979/£2453 to £7957/ £4905 (£22 586/QALY gained). Decreasing the probability of treatment uptake after testing positive for HBsAg from 0.42 to 0.1 increased the ICER to over £30 000 (£31 340/ QALY gained).

### Impact of variation in HBV prevalence and intervention impact (cost, effect and uptake)

Cost-effectiveness of HBV case-finding was strongly driven by HBV prevalence. Our sensitivity analyses indicated that the intervention would remain cost-effective under a £20 000 WTP threshold as long as HBV prevalence among the migrant population is equal to or exceeds 1% (figure 3).

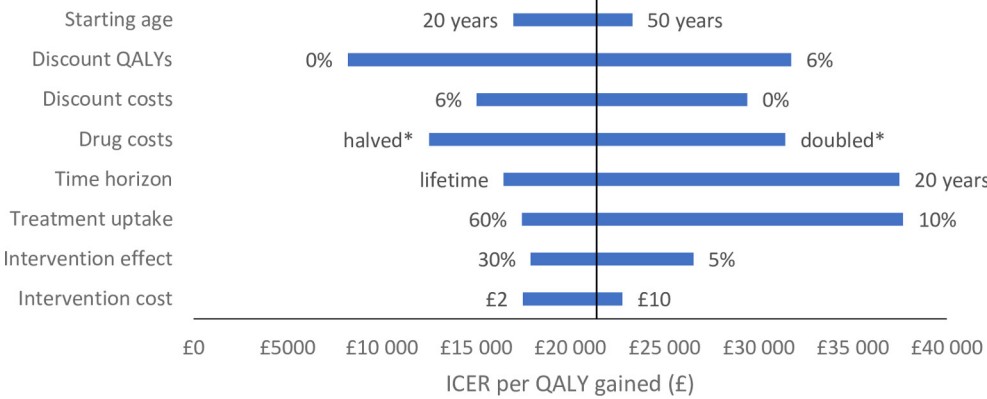

**Figure 2** Univariate sensitivity analysis on the ICER with a 2% HBV prevalence scenario. Y-axis indicates the base case ICER of £21 400 per QALY gained.HBV, hepatitis B virus; ICER, incremental cost-effectiveness ratio; QALY, quality-adjusted life-year. *Halves or doubles all baseline drug costs where relevant.

Due to the uncertainty in cost and intervention impact if scaled-up across the UK and among different migrant population, we additionally present a sensitivity analysis of the threshold HBV prevalence which would ensure that the intervention is cost-effective under a £20 000 WTP with varying costs and intervention effects (figure 4). The contour map shows that, for example, the intervention would be cost-effective at a prevalence of 1% if it cost £6 per person and the intervention effect was 20%. However, it would no longer be cost-effective at a 1% prevalence level and £6 cost if the intervention effect reduced to 10%.

## DISCUSSION

HBV case-finding using a one-time opt out approach in primary care settings has a high potential to be cost-effective among UK migrant populations with a HBV prevalence at or above an average of 1%. However, the results are sensitive to a number of factors including the intervention effect and cost, rate of treatment uptake, assuming a much shorter time horizon and (unrealistically) high discount rates and drug costs.

### Limitations

The main limitation with the analysis is the substantial uncertainty surrounding the costs of the intervention and its effect if this case-finding intervention were scaled-up to a national level. Nonetheless, extensive sensitivity analysis shows that the intervention remained cost-effective across a large range of evaluated scenarios. Thus, while establishing more robust estimates of the costs and effects of interventions to find cases of HBV will undoubtedly decrease the uncertainty around our results, we believe that the scope for the modelled intervention to be cost-effective is extremely high.

Current UK NHS HBV-testing policy is to contact household members once a case has been identified. However, we were unable to include this aspect in our analysis due to a lack of data specific to the target migrant populations on the size and age distribution of households of infected contacts, the probability that contacts were HBsAg+ and the likelihood that contacts could be traced in the first instance. The impact of excluding this process

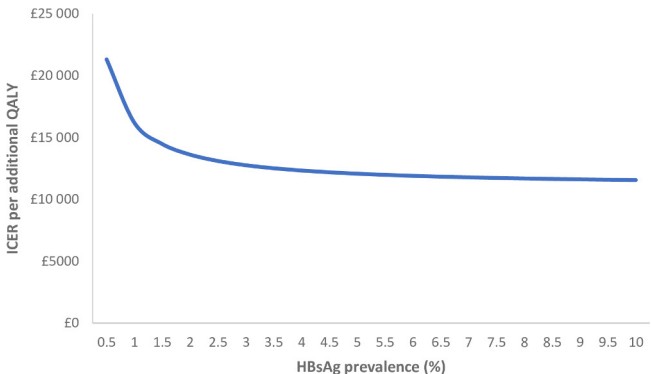

**Figure 3** Mean incremental cost-effectiveness ratio (ICER) of HBV screening by varying HBsAg prevalence. HBsAg, hepatitis B surface antigen; QALY, quality-adjusted life-year.

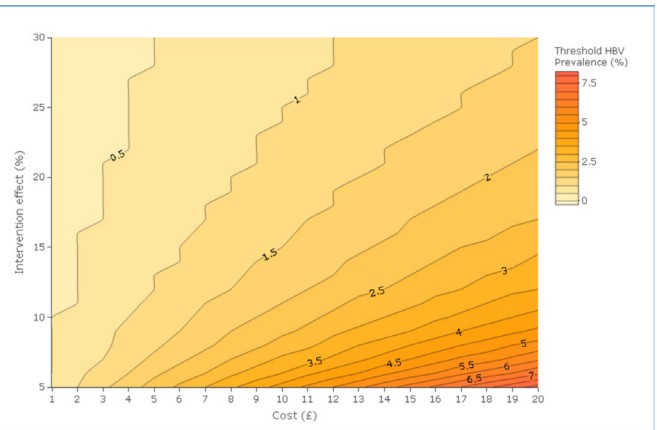

**Figure 4** Contour map showing for a range of costs (horizontal axis) and intervention effects (vertical axis), the threshold HBV prevalence (contours) where the intervention ICER falls under a £20 000 willingness to pay threshold. HBV, hepatitis B virus; ICER, incremental cost-effectiveness ratio.

on the ICER we report is difficult to determine. For example, if contact tracing results in a high proportion of people being treated for CHB, the ICER could decrease. Conversely, if many HBsAg- people are vaccinated against HBV, the ICER could increase as there is already evidence to suggest it is unlikely to be cost-effective.[15]

Finally, we did not model the possibility of simultaneously testing for HCV, which may increase the cost-effectiveness of the intervention though evidence on the HCV prevalence among migrants also has uncertainies.[19]

### Comparison with other studies

Five studies have examined the cost-effectiveness of screening for HBV among migrant populations. A Dutch study[17] found that screening migrants from countries with high or intermediate HBV prevalence (assuming a 3.4% chronic infection prevalence) was highly cost-effective (EUR9000 per QALY gained) at a screening campaign cost of approximately EUR11 per person eligible and 35% uptake—which is consistent with our sensitivity analysis. Another study explored screening and treatment of migrants from Asian and Pacific Islands in the USA,[40] finding it to be cost-effective (US$36 000 per QALY gained) but also assuming a much higher prevalence of HBV (10%), screening uptake (70%) and no screening programme costs aside from the diagnostic tests. Two studies examined the cost-effectiveness of screening all migrants to Canada,[41 42] both finding tenofovir-based treatment moderately cost-effective (CAD$40 000/QALY (~£22 000)) at 4.8%–6.5% chronic infection prevalence's. Our model assumes a lower prevalence of chronic HBV, higher treatment efficacy and lower treatment and screening costs than the North American studies, which may explain the difference in cost-effectiveness estimates. Finally, our results are partially consistent with findings from the recent HepFREE trial, which was found to be cost-effective (£8540/QALY) for a similar observed intervention effect (19.7% uptake of testing compared with 19.5% uptake in our study). However, HepFREE had higher intervention costs (>£25 per patient compared with £1–£20 in our model), combined HCV and HBV screening and identified patients on basis of ethnic group rather than country of birth.[43]

### CONCLUSIONS

Our analysis suggests that interventions to increase HBV case-finding in primary care among UK migrant populations with a prevalence of at least 1%—such as using a one-time opt out approach— could be cost-effective, underpinning current NIHC guidance.[44] Critically, at a threshold prevalence above 1%, this will encompass migrant populations from most countries with endemic HBV, even if there is a healthy migrant effect (with migrant populations in UK on average at lower risk than people in their country of origin[16]). These recent results support the recommendation that interventions to increase HBV case-finding in primary care among UK migrant populations should be expanded, but needs to be based on screening by country of birth rather than ethnic group.

**Contributors** AM, PV and MH designed the study. AM, AG, JW and NKM coded the analysis. All authors (AM, PV, MH, AG, JW, SK, MR and NKM) interpreted the data. AM and NKM wrote the first draft. All authors (AM, PV, MH, AG, JW, SK, MR and NKM) contributed to the manuscript drafting, approved of the final version and agreed to authorship.

**Funding** This work was originally funded by the UK National Institute for Health and Care Excellence. PV, MR and MH are affiliated with the National Institute for Health Research Health Protection Research Unit (NIHR HPRU) in Evaluation of Interventions at the University of Bristol in partnership with Public Health England (PHE). NKM, PV, and MH acknowledge funding from National Institute for Drug Abuse R01 DA037773. NKM also acknowledges funding from the University of California San Diego Center for AIDS Research (CFAR), a National Institute of Health (NIH) funded program [grant number P30 AI036214] , which is supported by the following NIH Institutes and Centers: NIAID, NCI, NIMH, NIDA, NICHD, NHLBI, NIA, NIGMS and NIDDK. The views in this publication are those of the authors and not necessarily those of the NHS, the National Institute for Health Research, the Department of Health and Social Care or Public Health England. AM, PV and JW are members of the NIHR's Sexually Transmitted Infections and Blood Borne Virus Health Protection Research Unit.

**Competing interests** NKM and PV have received unrestricted research grants from Gilead, outside the submitted work. NKM has received honoraria from Gilead and Merck. MH reports personal fees from Gilead, Abbvie and MSD.

**Patient consent for publication** Not required.

**Ethics approval** Ethical approval was not required for this study as it is an economic modelling exercise using published evidence and aggregate data from Public Health England .

**Provenance and peer review** Not commissioned; externally peer reviewed.

**Data sharing statement** Model code available on request to the corresponding author.

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
