## [Reviewer comments · BMJ Open]

ARTICLE DETAILS

TITLE (PROVISIONAL)	Chronic Hepatitis B virus case-finding in UK populations born abroad in intermediate or high endemicity countries: an economic evaluation
AUTHORS	Martin, Natasha; Vickerman, Peter; Khakoo, Salim; Ghosh, Anjan; Ramsay, Mary; Hickman, M; Williams, Jack; Miners, Alec

VERSION 1 - REVIEW

REVIEWER	Feng Yibin The University of Hong Kong Hong Kong
REVIEW RETURNED	04-Apr-2019

GENERAL COMMENTS	The paper aims to evaluate the cost-effectiveness of increased HBV case-finding among UK immigrants based on a one-time opt out case-finding approach in a primary care setting using a Markov approach. Authors reported their findings in compliance with Consolidated Health Economic Evaluation Reporting Standards (CHEERS) statement. The health economic evaluation requires a lot of work to analyze their data. However, the loose structure and vague reporting statement severely undermine the interpretation about the paper. This paper should be a lot of interest to the readership of the journal if authors can address the following issues: (1) The authors should clearly explain why the cost-effectiveness evaluation of increased HBV case-finding among UK immigrant population is necessary even existing HBV screening strategies are effective.(2) The targeted population should be well-defined. For example, what is "intermediate or high prevalence levels"? Why were Pakistani/British Pakistani people chosen as base case population?(3) Authors should introduce the Markov model more clearly, such as who and when will enter the model?(4) One of the limitations of this study is "substantial uncertainty surrounding the costs of the intervention and its effect if this case-finding intervention were scaled-up to a national level". So, why did authors not conduct two-way sensitivity analysis for intervention cost and effects to evaluate how their estimates would change and be scaled-up to a national level? Most important is that authors have to seek professional academic editing to present the introduction and analysis more accurately and concisely.
--

REVIEWER	Alec Morton University of Strathclyde, Glasgow, Scotland, UK
REVIEW RETURNED	12-Apr-2019

GENERAL COMMENTS	This review should be read with the background information that I have no particular knowledge of HepB. From a general modelling point of view I found the paper clearly written and explained. One question I had was whether HepB is asymptomatic up to the point where transplant is required, if so that would explain why patients do not present with the condition until the very late stage. If this is right, perhaps this could be made explicit. Another general comment is that the modelling draws heavily on third party studies. Most of this work is in the public domain but the work by PHE and HPA (refs 7 and 22) is not published in a journal. Is it or could be made available online and could references be provided? This would help ensure transparency and replicability. Other comments:  * p 6 - incentivized screening - would be good to explain this and say what is the magnitude of the incentive * p 8 - initial model cycle - how long is the initial model cycle? * fig 1 - there seem to be some stray punctuation marks * " and % in this diagram * p 9 "We assumed successful treatment..." this sentence is hard to understand, please rewrite * p10 – a different set of transition probabilities – what are they? * p14 – you say that you do a PSA but all the sensitivity analyses reported seem to be parametric SAs and there are no input distributions for the PSA
--

VERSION 1 – AUTHOR RESPONSE

Reviewer 1, comment 1: The authors should clearly explain why the cost-effectiveness evaluation of increased HBV case-finding among UK immigrant population is necessary even existing HBV screening strategies are effective.

Author reply: We thank the reviewer for this important point. Unfortunately existing HBV screening strategies among migrant populations are currently inadequate. As we state in the introduction, one UK study found that only 12% of migrants born in countries with intermediate to high endemicity had been tested for HBV. This indicates that the majority are likely to be unaware of their infection and at risk of liver disease progression. We underscore this point by expanding the introduction as below:

(Introduction) Although uncertain, it is also likely that a considerable number of people with chronic HBV remain undiagnosed. For example, in one study in Bristol only 12% of migrants born in countries with endemic prevalence >2% had been tested for HBV[9]. Due to the often asymptomatic nature of chronic infection, individuals with HBV infection can often remain undiagnosed until they develop advanced liver disease[10]. It is critical, therefore, that increased case-finding among UK migrant populations is enhanced to ensure timely treatment and follow-up to prevent complications from liver disease.

Reviewer 1, comment 2: The targeted population should be well-defined. For example, what is “intermediate or high prevalence levels”? Why were Pakistani/British Pakistani people chosen as base case population?

Author reply: We apologize this was unclear. The World Health Organization (Previsani N, Lavanchy D. 2002. Hepatitis B. Department of Communicable Diseases Surveillance and Response, World Health Organisation, Geneva) classification defines countries as intermediate prevalence (2-7%), and high prevalence (>8%). We now add this information to the introduction and methods. Additionally, we note that although we base our analysis on data from a study among Pakistani individuals in London, our targeted population is individuals born in countries with intermediate or high prevalence levels ($\geq 2\%$), so we evaluate the impact of this intervention in populations with a range of HBV prevalences as observed among UK migrants born from these countries. We add additional explanation in the methods as below.

(Title) Chronic Hepatitis B virus case-finding in UK populations born abroad in intermediate or high endemicity countries: an economic evaluation

(Abstract) Objectives: The majority (>90%) of new or undiagnosed cases of hepatitis B virus (HBV) in the United Kingdom (UK) are among individuals born in countries with intermediate or high prevalence levels ($\geq 2\%$).

(Abstract) Intervention: HCV case-finding among UK migrant populations born in countries with intermediate or high prevalence levels ($\geq 2\%$) in a primary care setting compared to no intervention (background testing).

(Introduction) The vast majority (80% to 90%) of newly diagnosed chronic HBV infections are among migrant individuals living in the UK that were born overseas in countries with intermediate (2-7%) or high HBV prevalence ($\geq 8\%$) as defined by the World Health Organization, such as China or Pakistan

(Methods) The economic evaluation was undertaken using a Markov approach, where a closed cohort of UK individuals born in countries with intermediate or high prevalence levels ($\geq 2\%$) move between a set of discrete health states representing HBV infection stage.

(Methods) Our study evaluates the cost-effectiveness of HBV case-finding in the U.K. for individuals born in countries with intermediate or high prevalence levels ($\geq 2\%$). The base case analysis uses the results from an uncontrolled study in which Pakistani/British Pakistani people registered at general practices (GPs) in London’s East End were written to and invited to ‘opt out’ of being tested for hepatitis B and C infection. Those who did not opt out were telephoned and asked to attend a clinic for testing[19] The intervention was designed to increase the likelihood of testing for each infection, assumed in this analysis to occur over the initial model cycle of one year. After this time, the

intervention effect was assumed to be zero, with the probability of testing reverting to background levels. The comparator programme or 'no intervention' was defined as the background likelihood of testing through existing routes such as sexual health or genitourinary medicine clinics, antenatal clinics or primary care[24]. Although we base our analysis on data from a study among Pakistani/British Pakistani individuals in London, we evaluate the potential impact of this intervention in populations with a range of HBV prevalences as observed among UK migrants born in countries with intermediate or high prevalence levels ($\geq 2\%$).

Reviewer 1, comment 3: Authors should introduce the Markov model more clearly, such as who and when will enter the model?

Author reply: We apologize for the confusion and have added further details in the initial description of the Markov model to better clarify that it simulates a closed cohort of UK migrants born in countries with intermediate or high prevalence levels.

(Methods) The economic evaluation was undertaken using a Markov approach, where a closed cohort of UK individuals born in countries with intermediate or high prevalence levels ($\geq 2\%$) move between a set of discrete health states representing HBV infection stage.

Reviewer 1, comment 4: One of the limitations of this study is "substantial uncertainty surrounding the costs of the intervention and its effect if this case-finding intervention were scaled-up to a national level". So, why did authors not conduct two-way sensitivity analysis for intervention cost and effects to evaluate how their estimates would change and be scaled-up to a national level?

Author reply: We apologise if the results were unclear as we have conducted the analysis suggested by the reviewer (see Figure 4). In the sensitivity analysis we provide the information that the reviewer suggests- specifically, a two-way sensitivity analysis with varying cost and intervention effects, showing the threshold HBV prevalence that would ensure the intervention is cost-effective.

Reviewer 1, comment 5: Most important is that authors have to seek professional academic editing to present the introduction and analysis more accurately and concisely.

Author reply: We thank the reviewer for this suggestion and have edited the introduction and analysis throughout to improve clarity and precision. We welcome any additional specific suggestions.

Reviewer 2, comment 1: One question I had was whether HepB is asymptomatic up to the point where transplant is required, if so that would explain why patients do not present with the condition until the very late stage. If this is right, perhaps this could be made explicit.

Author reply: We thank the reviewer for this question and now clarify the asymptomatic nature of the disease in the introduction. We also note that we explicitly discuss this in the methods where we state

that "Due to the severity of the disease and likely presentation, the infection status of all individuals with CHB was assumed to become known when they developed DC, HCC or required a liver transplant."

(Introduction) Due to the often asymptomatic nature of chronic infection[10], individuals with HBV infection can often remain undiagnosed until they develop advanced liver disease.

Reviewer 2, comment 2: Another general comment is that the modelling draws heavily on third party studies. Most of this work is in the public domain but the work by PHE and HPA (refs 7 and 22) is not published in a journal. Is it or could be made available online and could references be provided? This would help ensure transparency and replicability.

Author reply: We note that the refs 7 and 22 are in the published domain as online reports; we have edited these references to include web addresses and thank the reviewer for noticing this.

Reviewer 2, comment 3: p 6 - incentivized screening - would be good to explain this and say what is the magnitude of the incentive

Author reply: The following text has been added for clarification:

(Discussion) This changed earlier this year when the results of a randomized controlled trial (HepFREE) showed that incentivized screening of HBV and HCV in first and second-generation migrants in a primary care setting was shown to be effective and cost-effective in the UK; the incentive included a startup payment of £500 per general practice, £25 for each enrolled participant and support from a dedicated clinician 3 days a week

Reviewer 2, comment 4: p 8 - initial model cycle - how long is the initial model cycle?

Author reply: We now clarify that the initial model cycle is one year.

(Methods) The intervention was designed to increase the likelihood of testing for each infection, assumed in this analysis to occur over the initial model cycle of one year.

Reviewer 2, comment 5: fig 1 - there seem to be some stray punctuation marks * " and % in this diagram

Author reply: These punctuation marks are superscripts to represent the footnote, as described in the figure legend.

Reviewer 2, comment 6: p 9 "We assumed successful treatment..." this sentence is hard to understand, please rewrite

Author reply: We apologize for the confusion and have revised the sentence as below for clarity.

(Methods) Individuals who had raised ALT and HBV (active) levels and who were CHB HBeAg+ were assessed for fibrosis and offered treatment with pegylated interferon for the first year, followed by tenofovir until seroconversion is achieved (as per NICE guidelines) or later stage CHB developed. We assumed successful treatment of these individuals resulted in normalization of ALT and lowering of HBV DNA levels, therefore resulting in transition to the HBeAg seroconverted stage.

Reviewer 2, comment 7: p10 – a different set of transition probabilities – what are they?

Author reply: We report these probabilities in Supplementary Tables 1 and 2, and now have edited text to reference these tables for clarity.

(Methods) Individuals with CHB whose infection status was unknown and those that tested HBsAg+, but declined treatment, were assumed to develop progressive disease according to a set of defined transition probabilities, with different probabilities used for those who accepted treatment (Supplementary Tables 1 and 2).

Reviewer 2, comment 8: p14 – you say that you do a PSA but all the sensitivity analyses reported seem to be parametric SAs and there are no input distributions for the PSA

Author reply: We apologize for the confusion. We implement a PSA for the main analysis, with the input distributions for the PSA reported in Table 1 and Supplementary Tables 1-3. The PSA is reported in the main results as follows: "ICER of £13,625 per QALY gained (95% credible interval £7,121 to £27,588) The intervention was 87% and 98% likely to be cost-effective at £20,000 and £30,000 WTP per additional QALY thresholds, respectively." In addition to this multivariate probabilistic analysis, we additionally incorporate numerous one-way sensitivity analysis by varying single parameters. We clarify this in the methods by incorporating a new section: "Main outcomes"

(Methods) Main outcomes: Our main results incorporate a probabilistic sensitivity analysis (PSA), in which relevant parameters are simultaneously sampled 5,000 times to represent underlying uncertainty, including the costs, utilities probabilities and disease progression parameters. We present total and incremental costs, QALYs, and incremental cost-effectiveness ratios (ICERs). Mean and

2.5-97.5% centile (95% CI) results are presented. We additionally present the proportion of simulations which are cost-effective under £20,000 and £30,000 WTP thresholds.

VERSION 2 – REVIEW

REVIEWER	Feng Yibin The University of Hong Kong Hong Kong
REVIEW RETURNED	21-May-2019

GENERAL COMMENTS	The authors have addressed all of my comments.
--

REVIEWER	Alec Morton University of Strathclyde, UK
REVIEW RETURNED	06-May-2019

GENERAL COMMENTS	I think my comments have been adequately addressed
--